**Data Availability Statement:** All the data related to this research article are available in this manuscript as text, tables and figures. However, to replicate

# Changing prevalence and factors associated with female genital mutilation in Ethiopia: Data from the 2000, 2005 and 2016 national demographic health surveys

Gedion Asnake Azeze[1]*, Anita Williams[2], Hannock Tweya[3,4], Mohammed Suleiman Obsa[5], Taklu Marama Mokonnon[1], Zewde Zema Kanche[6], Robera Olana Fite[7], Anthony D. Harries[3,8]

1 Department of Midwifery, College of Health Science and Medicine, Wolaita Sodo University, Sodo, Ethiopia, 2 Medical Department, Luxembourg Operational Research Unit (LuxOR), Médecins Sans Frontières - Operational Centre Brussels, Luxembourg, Luxembourg, 3 International Union against Tuberculosis and Lung Disease, Paris, France, 4 The Lighthouse Clinic, Lilongwe, Malawi, 5 Department of Anesthesia, College of Health Science and Medicine, Wolaita Sodo University, Sodo, Ethiopia, 6 Department of Pharmacy, College of Health Science and Medicine Wolaita Sodo University, Sodo, Ethiopia, 7 Department of Nursing, College of Health Science and Medicine, Wolaita Sodo University, Sodo, Ethiopia, 8 London School of Hygiene and Tropical Medicine, London, United Kingdom

* dearged2011@gmail.com

## Abstract

### Setting

Female genital mutilation (FGM) is a traditional surgical modification of the female genitalia comprising all procedures involving partial or total removal of the external female genitalia or other injury to the female genital organs for cultural or nontherapeutic reasons. It can be harmful and violates girls' and women's human rights. FGM is a worldwide problem but mainly practiced in Africa. FGM is still widely practiced in Ethiopia despite being made a criminal offence in 2004.

### Objective

Using data from three Ethiopian Demographic Health Surveys (EDHS) conducted in 2000, 2005 and 2016 the objective was to assess changes in prevalence of FGM and associated factors among women of reproductive age and their daughters.

### Methods

EDHS datasets for the three surveys included data on FGM prevalence and socio-demographic factors. After weighting, the data were analysed using frequencies, proportions and the chi square test for trend. Categorical variables associated with FGM in 2016 were compared using OpenEpi and presented as prevalence ratios (Pr) with 95% Confidence Intervals (CI). Levels of significance were set at 5% ($P<0.05$).

our study findings, the EDHS datasets are in the public domain on the DHS measure survey web site which is available at: https://dhsprogram.com/data/available-datasets.cfm (accessed 16 June 2020). The authors had no special access privileges to the DHS data, and other researchers will be able to access the data in the same manner as the authors using the provided URL link. Data access requests may also be sent to Bridgette Wellington (Data Archivist at The Demographic and Health Surveys (DHS) Program) at E-mail address: archive@dhsprogram.com.

**Funding:** Funding was from the United Kingdom's Department for International Development (DFID) and La Fondation Veuve Emile Metz-Tesch. The funders had no role in study design, data collection and analysis, decision to publish or preparation of the manuscript.

**Competing interests:** The authors have declared that no competing interests exist.

**Abbreviations:** EDHS, Ethiopian Demographic and Health Survey; FGM, Female Genital Mutilation; ICF, Inner City Fund International; SNNPR, Southern Nations Nationalities and Peoples Region; UNFPA, United Nations Population Fund; UNICEF, United Nations Children's Fund; WHO, World Health Organization.

## Results

There was overall decline in FGM prevalence (from 79.9% to 74.3% to 65.2%, P<0.001), especially in younger women aged 15-19 years, and in the proportion of women who believed that the practice should continue (from 59.7% to 28.3% to 17.5%, P<0.001). There was also a decreasing trend of FGM in the daughters of the mothers who were interviewed, with prevalence significantly lower in mothers who had not themselves undergone FGM. Most (88.3%) women with FGM had the surgery as a child with the procedure mainly performed by a traditional circumciser (87.3%). Factors associated with higher FGM prevalence and lack of progress over the sixteen years included living in certain regions, especially Somali where FGM prevalence remained consistently >95%, lack of school education, coming from rural areas and living in less wealthy households.

## Conclusion

Although progress has been slow, the prevalence of FGM in Ethiopia has declined over time. Recommendations to quicken the trajectory of decline targeting integrated interventions to high prevalence areas focusing on mothers, fathers, youngsters, religious leaders and schools and ensuring that all girls receive some form of education.

## Introduction

Female genital mutilation (FGM), also known as female circumcision or female genital cutting, is a traditional surgical modification of the female genitalia comprising all procedures involving partial or total removal of the external female genitalia or other injury to the female genital organs for cultural or nontherapeutic reasons [1]. There are four main classifications of FGM [1]: in brief, these are Type I (Clitoridectomy—partial or total); Type II (Excision—partial or total removal of the clitoris and the labia minor); Type III (Infibulation—narrowing of the vaginal orifice with creation of a covering seal by cutting and apposition of the labia minor and/or labia majora, with or without excision of the clitoris); Type IV (all other harmful procedures to the female genitalia for non-medical purposes, e.g., pricking, piercing, incising, scraping and cauterization). Types I and II are the most common forms while Type III (Infibulation) is the most severe form accounting for 10% of cases globally [2]. The practice of FGM has severe harmful consequences, such as urinary tract infections, bacterial vaginosis, dyspareunia and obstetric complications which are more common in Type III FGM compared with Types I and II FGM [3]. The practice of FGM is also internationally recognized as a violation of girls' and women's human rights [1, 2].

Despite this, FGM is a worldwide problem. More than 200 million girls and women globally are living with the effects of FGM, of whom 44 million are under 15 years of age [2, 4]. FGM is mainly practiced in Africa, often by non-medical practitioners [5]. Reasons for its continuation, especially in Africa, are multifactorial and include social, cultural, traditional and/or religious beliefs [2, 5]. These encompass beliefs about female cleanliness, cultural identity, protection of virginity, prevention of immorality, better prospects for marriage and a necessary ritual for initiation into womanhood. The procedure is often carried out between the aged of six and eight years.

he first international response to address the problem of FGM was a joint statement in 1997 made by the World Health Organization (WHO) in conjunction with the United Nations

Children's Fund (UNICEF) and the United Nations Population Fund (UNFPA). In 2017, the United Nations General Assembly reaffirmed the resolution to eliminate FGM [6]. These combined international efforts and accompanying legal frameworks have resulted in a decreasing prevalence of FGM, although progress towards eradicating the practice is too slow and has not been uniform [7].

FGM has been widely practiced in Ethiopia for years where it is regarded as a major public health problem and an important cause of maternal mortality and adverse birth outcomes [7–9]. In 2004, the Government of Ethiopia passed a law making it a criminal offence to perform or procure FGM in the country [10]. An Ethiopian Demographic and Health Survey (EDHS) conducted one year later in 2005 estimated that the prevalence of FGM was 74% amongst women aged 15-49 years and 38% amongst their daughters, although prevalence rates varied hugely amongst different regions from 29% and 27% in Tigray and Gambela regions respectively to 82% in Harari and 97% in the Somali regions [11].

Ethiopia has conducted regular EDHS in 2000, 2005, 2011 and 2016. In all of these surveys except for 2011, standardized questions were asked about characteristics of FGM and demographic information was also collected. Data from the first two surveys (2000 and 2005) were used to assess geographical variation and factors associated with FGM prevalence in the country. The findings showed that the weighted prevalence of FGM had decreased over the five years with risk factors for increased prevalence being wealth, increasing age, rural residence and having Islamic faith and protective factors being maternal education [12].

Recommendations have been made to improve data collection on FGM [2]. Self-reported data on whether FGM has taken place has been found to be reliable, although there tends to be under-reporting of the severity of the procedure [13]. Despite these limitations and with the publication of the 2016 EDHS, we felt it would be valuable to look at the self-reported data in the three EDHS to assess whether there had been any significant changes in FGM in reproductive age women and their daughters over the three time periods in the country (2000, 2005 and 2016).

The aim of the study therefore was to document changes in prevalence and factors associated with FGM among women of reproductive age (15-49 years) and their daughters from the datasets of the 2000, 2005 and 2016 EDHS. Specific objectives were to examine: i) changes in prevalence, opinions and self-reported practices of FGM in women who were interviewed; ii) socio-demographic factors associated with women who had FGM; and iii) prevalence of FGM among daughters as reported by their mothers.

## Materials and methods

### Study design

This was a secondary analysis of cross-sectional studies done in the three EDHS.

### Setting

**General setting.** Ethiopia is located in the Horn of Africa and is the second most populated country in sub-Saharan Africa with almost 105 million inhabitants [14]. Life expectancy at birth is 66 years [14], whilst the GDP per capita is at USD$735 [15]. Administratively the country is divided into nine geographical regions and two administrative cities, and there are approximately 80 different ethnic groups [16]. There is a shortage of skilled health care professionals in the country, estimated at 2.8 healthcare workers per 10,000 population, and health service coverage for reproductive age women, infants and children is estimated at 59% [17]. Traditional practices, including FGM, are common and are used by 80% or more of the population, and female literacy in the country is estimated at 29% [16]. These factors all contribute

to a high maternal mortality ratio of 446 per 100,000 live births and an under-five mortality rate of 60 per 1,000 live births [17].

**Specific setting (Ethiopian Demographic and Health Surveys).** Demographic and Health Surveys are used in most countries of the world to collect data on marriage, fertility, family planning, reproductive health, child health and HIV/AIDS so that decision makers in participating countries have improved information and analyses useful for informed policy choices [18]. The 2000 [19], 2005 [11] and 2016 [16] EDHS used the same methodology, inter-viewing a sample that represented the population at the national and regional levels, and for urban and rural areas. Eligible participants were women aged 15-49 years and men aged 15-59 years in randomly selected households across Ethiopia. In 2000, 2005 and 2016, the numbers of households selected were 14,642, 14,500 and 16,650 respectively and the numbers of female participants were 15,367 (2000), 14,070 (2005), and 15,683 (2016). The EDHS protocols were reviewed and approved by the Federal Democratic Republic of Ethiopia Ministry of Science and Technology and Institutional Review Board of the Inner City Fund (ICF) International [11, 16, 19]. For the purpose of this study, detailed information in the EDHS was collected on a wide range of socio-demographic factors and characteristics of FGM.

## Study population

The study population included all women of reproductive age (15-49 years) who were inter-viewed about themselves and their daughters in the 2000, 2005 and 2016 EDHS.

## Data variables, source of data and data collection

Data variables in our analysis included: year of EDHS; number of reproductive age women aged 15-49 years completing the survey questionnaire; number of women interviewed who had ever been circumcised; number whose genital area had been sewn closed; and whether they thought circumcision should be continued or stopped; for 2016 the age at the time of cir-cumcision and who performed the circumcision. Socio-demographic factors included: age at the time of interview; region of residence; education; religion; urban or rural residence; occu-pation and wealth index. Wealth index was calculated for each household based on the number and kind of common goods (for example, a television) that were owned and housing charac-teristics such as source of drinking water, toilet facilities and floor materials [16]. For daughters of mothers who were interviewed, data variables included: whether mothers had been circum-cised or not; total number of daughters; and number of daughters who were circumcised.

The sources of data were the completed EDHS questionnaires, which were extracted as .sav files for SPSS (version 25, IBM, New York, USA). The data were extracted between February and April, 2019.

## Analysis and statistics

The three data sets from EDHS were weighted before analysis to compensate for unequal prob-ability of selection among geographic strata to restore the representativeness of the sample using standard DHS methodology [20]. After weighting, data were analysed using OpenEpi (Open Source Epidemiological Statistics for Public Health, version 3.03) and SPSS (version 25, IBM, New York, USA). Frequencies and proportions for each variable were calculated and described. The proportions of women with FGM and factors associated with FGM in each of the three time periods were compared with the chi square test using OpenEpi. Categorical var-iables associated with FGM in 2016 were compared using OpenEpi and presented as preva-lence ratios (Pr) with 95% confidence intervals (CI). Levels of significance were set at 5% ($P<0.05$).

### Ethics

Permission to access and analyse the EDHS files was obtained from the Demographic and Health Survey Program, Inner City Fund (ICF), Virginia, USA. Local ethics approval was obtained from the Chief Academic and Research Directorate Office, College of Health Sciences, Wolaita Sodo University, Sodo, Ethiopia. Ethics approval was also obtained from The Ethics Advisory Group, International Union Against Tuberculosis and Lung Disease, Paris, France (EAG Number 20/19). As the data used in this study was all secondary and in the public domain, individual consent is not required.

## Results

### Changing prevalence, opinions and self-reported practices of FGM in 2000, 2005 and 2016

The numbers and proportions of women interviewed in 2000, 2005 and 2016 who had undergone FGM are shown in Fig 1. Over the three time periods, there was a significant decline in the proportion of women reportedly having FGM from 79.9% to 74.3% to 65.2% (differences between 2000-2005 and between 2005-2016 = P<0.001). Opinions about whether the practice of FGM should continue or not changed significantly over the three time periods, with fewer women agreeing that FGM should continue and more women agreeing that it should stop (Table 1). In women who had undergone FGM, a small but increasing proportion had the most severe form with the genital area sown closed or "infibulation" (Table 2). Characteristics associated with FGM in women in 2016 are shown in Table 3. FGM was performed as a child aged 14 years or younger in 88.3% of women and was performed in nearly half of these women when they were under five years of age. The FGM surgery was mainly performed by traditional circumcisers in 87.3%.

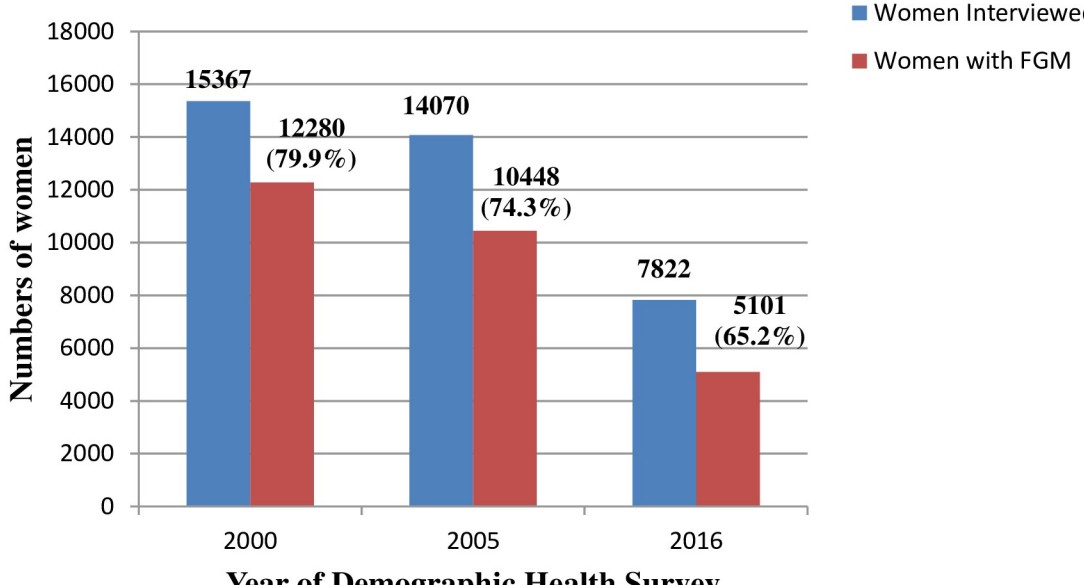

**Fig 1. Women in Ethiopia who were interviewed and who reported female genital mutilation (FGM) in 2000, 2005 and 2016.**

**Table 1. Opinions about whether the practice of female genital mutilation (FGM) in Ethiopia should continue: 2000, 2005 and 2016.**

| Characteristic | 2000 | | 2005 | | 2016 | |
|---|---|---|---|---|---|---|
| | N | (%, 95% CI) | N | (%, 95% CI) | N | (%, 95% CI) |
| Total | 15367 | | 14070 [a] | | 7822 | |
| Should the practice of FGM continue? | | | | | | |
| Yes | 9179 | (59.7, 58.9-60.5) | 3981 | (28.3, 27.6-29.0) [aa] | 1277 | (16.3, 15.5-17.2) [bb] |
| No | 4110 | (26.7, 26.1-27.5) | 7893 | (56.1, 55.3-56.9) [aa] | 6137 | (78.5, 77.5-79.4) [bb] |
| Uncertain | 826 | (5.4, 5.0-5.7) | 659 | (4.7, 4.3-5.0) [a] | 408 | (5.2, 4.7-5.7) |
| Missing data | 1252 | (8.2, 7.7-8.6) | 1537 | (10.9, 10.4-11.5) [aa] | 0 | 0 |

FGM = female genital mutilation; 95%CI = 95% confidence intervals.

All data were weighted according to DHS guidance [20].

[aa] = P<0.001;

[a] = P<0.05—proportions in 2005 compared with proportions in 2000 using the chi square test.

[bb] = P<0.001—proportions in 2016 compared with proportions in 2005 using the chi square test.

## Socio-demographic factors associated with FGM in 2000, 2005 and 2016

Factors associated with changing prevalence of FGM are shown in Table 4. Between 2000 and 2005 there was a significant decline in FGM prevalence in the younger and older age groups, while between 2005 and 2016 there was a significant decline in FGM prevalence in all age groups except for those aged 30-34years. The greatest change was observed in the age group 15-19 where FGM prevalence decreased from 70.7% in 2000, to 64.2% in 2005 to 47.1% in 2016. Over the two time periods, regions showing a significant decline in FGM were Tigray, SNNPR, Amhara, Addis Ababa, Oromiya, Dire Dawa and Affar. The Somali region had a high FGM prevalence throughout. Amongst women who had no education there was a significant change in FGM prevalence between 2005 and 2016, but in women who received any education there was a significant decline in FGM prevalence over the two time periods. FGM was highly prevalent amongst Muslim women, but all religious groups showed a significant decline in FGM either in the first time period or the second time period. Women in both urban and rural settings had a significant decline in FGM, although prevalence was always higher in those

**Table 2. Severity of female genital mutilation (FGM) in Ethiopian women: 2000, 2005 and 2016.**

| Characteristics | 2000 | | 2005 | | 2016 | |
|---|---|---|---|---|---|---|
| | Women with FGM | | Women with FGM | | Women with FGM | |
| | N | (%, 95%CI) | N | (%, 95%CI) | N | (%, 95%CI) |
| All women with FGM | 12280 | | 10448 | | 5101 | |
| FGM sewn closed | 369 | (3.0, 2.7-3.3) | 638 | (6.1, 5.7-6.6) [aa] | 334 | (6.5, 5.9-7.3) |
| FGM not sewn closed | 11747 | (95.6, 95.3-96.0) | 8996 | (86.1, 85.4-86.8) [aa] | 4429 | (86.8, 85.9-87.7) |
| Uncertain about severity of FGM | 104 | (0.8, 0.7-1.0) | 529 | (5.1, 4.7-5.5) [aa] | 338 | (6.7, 6.1-7.3) [bb] |
| Missing data | 60 | (0.6, 0.4-0.6) | 285 | (2.7, 2.4-3.1) [aa] | 0 | |

FGM = female genital mutilation; 95%CI = 95% confidence intervals.

All data were weighted according to DHS guidance [20].

[aa] = P<0.001;

[a] = P<0.05—proportions in 2005 compared with proportions in 2000 using the chi square test.

[bb] = P<0.001—proportions in 2016 compared with proportions in 2005 using the chi square test.

**Table 3. Characteristics of women with female genital mutilation (FGM) in Ethiopia in 2016.**

| Characteristics | N | (%, 95%CI) |
|---|---:|---|
| **All women with FGM** | **5101** | |
| **Age in years at which FGM was performed**: | | |
| Less than 5 | 2478 | (48.6, 47.2-50.0) |
| 5-9 | 1108 | (21.7, 20.6-22.9) |
| 10-14 | 918 | (18.0, 17.0-19.1) |
| 15 and above | 301 | (5.9, 5.3-6.6) |
| Uncertain | 296 | (5.8, 5.2-6.5) |
| **FGM performed by**: | | |
| Traditional circumciser | 4453 | (87.3, 86.4-88.2) |
| Traditional birth attendant | 133 | (2.6, 2.2-3.1) |
| Other traditional practitioner | 11 | (0.2, 0.1-0.4)) |
| Medical doctor | 4 | (0.1, 0.02-0.2) |
| Trained nurse/mid-wife | 28 | (0.5, 0.4-0.8) |
| Other health professional | 19 | (0.4, 0.2-0.6) |
| Uncertain | 453 | (8.9, 8.1-9.7) |

FGM = female genital mutilation; 95%CI = 95% confidence intervals.

All data were weighted according to DHS guidance [20].

who lived in rural areas. For all occupational groups there was a significant decline in FGM prevalence.

Prevalence ratios for FGM in women interviewed in 2016 are shown in Table 5. Compared with Tigray, women from all regions had a significantly higher prevalence of FGM Women in the regions of Harari, Affar and Somali had prevalence ratios three to four times higher than women in Tigray. Compared to women with no education, FGM prevalence was significantly lower for women who attended primary, secondary and tertiary level education. Compared with traditional religious beliefs women of muslim faith had a higher prevalence of FGM. There was significantly higher FGM prevalence in rural compared with urban areas. Women engaged in manual and other occupations had a lower prevalence of FGM compared with those who were unemployed. Finally, women with poorer household wealth had higher FGM prevalence compared with women from the richest households.

## Changing prevalence of FGM among daughters as reported by their mothers

Changing prevalence of FGM among daughters of the female participants over the study period is shown in Figs 2 and 3. In daughters born to mothers who had FGM themselves, the prevalence of FGM significantly declined from 56.2%, to 45.6% to 16.7% ($P<0.001$) (Fig 2). In daughters born to mothers who did not have FGM, the overall prevalence of FGM was significantly lower and also significantly declined from 10.1%, to 8.3% to 1.7% ($P<0.001$) (Fig 3).

## Discussion

This is the first study from Ethiopia using data from three EDHS to document what had happened with FGM in the country over a 16-year period. There were some important findings.

First, and on an encouraging note, FGM became less common over time. The overall prevalence of FGM significantly decreased, with the decline particularly marked in young women

**Table 4. Socio-demographic factors associated with female genital mutilation (FGM) in Ethiopia: 2000, 2005 and 2016.**

| Characteristics | 2000 | | | 2005 | | | 2016 | | |
|---|---|---|---|---|---|---|---|---|---|
| | Interviewed | With FGM | | Interviewed | With FGM | | Interviewed | With FGM | |
| | N | N | (%) | N | N | (%) | N | N | (%) |
| Total | 15367 | 12280 | (79.9) | 13683 [α] | 10448 | (76.3) [aa] | 7822 | 5101 | (65.2) [bb] |
| **Age at interview in years:** | | | | | | | | | |
| 15-19 | 3710 | 2624 | (70.7) | 3160 | 2029 | (64.2) [aa] | 1670 | 786 | (47.1) [bb] |
| 20-24 | 2859 | 2240 | (78.3) | 2471 | 1860 | (75.3) [a] | 1291 | 756 | (58.6) [bb] |
| 25-29 | 2585 | 2105 | (81.4) | 2455 | 1954 | (79.6) | 1474 | 996 | (67.6) [bb] |
| 30-34 | 1841 | 1586 | (86.1) | 1772 | 1410 | (79.6) [aa] | 1201 | 924 | (76.9) |
| 35-39 | 1716 | 1434 | (83.6) | 1571 | 1302 | (82.9) | 1016 | 758 | (74.6) [bb] |
| 40-44 | 1392 | 1195 | (85.8) | 1150 | 969 | (84.3) | 649 | 471 | (72.6) [bb] |
| 45-49 | 1264 | 1097 | (86.8) | 1104 | 924 | (83.7) [a] | 521 | 410 | (78.7) [b] |
| **Region:** | | | | | | | | | |
| Tigray | 970 | 346 | (35.7) | 821 | 269 | (32.8) | 540 | 131 | (24.2) [bb] |
| Gambela | 40 | 17 | (42.5) | 44 | 12 | (27.3) | 21 | 7 | (33.3) |
| SNNPR | 3285 | 2414 | (73.5) | 2982 | 2127 | (71.3) | 1652 | 1024 | (61.9) [bb] |
| Benshagul-Gumuz | 160 | 118 | (73.8) | 122 | 84 | (68.9) | 75 | 47 | (62.7) |
| Amhara | 3820 | 3045 | (79.7) | 3288 | 2386 | (72.6) [aa] | 1827 | 1127 | (61.7) [bb] |
| Addis Ababa | 684 | 546 | (79.8) | 700 | 497 | (71.0) [aa] | 466 | 251 | (53.9) [bb] |
| Oromiya | 5937 | 5332 | (89.8) | 4989 | 4369 | (87.6) | 2881 | 2178 | (75.6) [bb] |
| Dire Dawa | 79 | 75 | (94.9) | 69 | 64 | (92.8) | 47 | 35 | (74.5) [b] |
| Harari | 40 | 38 | (95.0) | 38 | 33 | (86.8) | 18 | 15 | (83.3) |
| Affar | 178 | 175 | (98.3) | 144 | 134 | (93.1) [a] | 67 | 61 | (91.0) |
| Somali | 174 | 174 | (100) | 486 | 473 | (97.3) | 228 | 225 | (98.7) |
| **Education:** | | | | | | | | | |
| None | 11551 | 9291 | (80.4) | 9044 | 7165 | (79.2) | 3787 | 2758 | (72.8) [bb] |
| Primary level | 2425 | 1902 | (78.4) | 3045 | 2211 | (72.6) [aa] | 2679 | 1662 | (62.0) [bb] |
| Secondary level | 1304 | 1013 | (77.7) | 1408 | 964 | (68.4) [aa] | 907 | 453 | (49.9) [bb] |
| Tertiary level | 87 | 74 | (85.1) | 186 | 108 | (58.1) [aa] | 450 | 228 | (50.7) |
| **Religion:** | | | | | | | | | |
| Traditional | 503 | 336 | (66.8) | 189 | 93 | (49.2) [aa] | 61 | 33 | (54.8) |
| Catholic | 175 | 117 | (66.9) | 171 | 133 | (77.8) [a] | 66 | 39 | (58.2) [b] |
| Protestant | 2432 | 1748 | (71.9) | 2642 | 1891 | (71.6) | 1862 | 1226 | (65.8) [bb] |
| Orthodox Christian | 7764 | 5961 | (76.8) | 6608 | 4672 | (70.7) [aa] | 3424 | 1856 | (54.2) [bb] |
| Muslim | 4456 | 4091 | (91.8) | 3950 | 3566 | (90.3) | 2363 | 1942 | (82.2) [bb] |
| Others | 37 | 27 | (73.0) | 123 | 91 | (74.0) | 46 | 5 | (10.8) [bb] |
| **Residence:** | | | | | | | | | |
| Urban | 2791 | 2227 | (79.78) | 2362 | 1713 | (72.5) [aa] | 1714 | 924 | (53.9) [bb] |
| Rural | 12576 | 10053 | (79.9) | 11321 | 8735 | (77.1) [aa] | 6108 | 4177 | (68.4) [bb] |
| **Occupation:** [β] | | | | | | | | | |
| Unemployed | 5657 | 4501 | (79.6) | 9030 | 6874 | (76.1) [aa] | 3839 | 2564 | (66.8) [bb] |
| Professional | 2429 | 2032 | (83.7) | 1697 | 1280 | (75.4) [aa] | 1729 | 1138 | (65.8) [bb] |
| Agricultural | 5691 | 4417 | (77.6) | 2412 | 1938 | (80.3) [a] | 1629 | 1055 | (64.8) [bb] |

(*Continued*)

**Table 4.** (Continued)

| Characteristics | 2000 | | | 2005 | | | 2016 | | |
|---|---|---|---|---|---|---|---|---|---|
| | Interviewed | With FGM | | Interviewed | With FGM | | Interviewed | With FGM | |
| | N | N | (%) | N | N | (%) | N | N | (%) |
| Manual | 1576 | 1318 | (83.6) | 500 | 328 | (65.6) [aa] | 441 | 242 | (55.4) [bb] |

[α] this number excludes 387 women who either responded as "do not know" or had "missing data" for socio-demographic variables.

[β] for 2005 and 2016, there were 44 and 184 women who responded "don't know" and "other" with respect to occupation respectively.

FGM = female genital mutilation; SNNPR = Southern Nations Nationalities and Peoples Region; 95%CI = 95% confidence intervals.

All data were weighted according to DHS guidance [20].

[aa] = P<0.001;

[a] = P<0.05—proportions in 2005 compared with proportions in 2000 using the chi square test.

[bb] = P<0.001;

[b] = P<0.05—proportions in 2016 compared with proportions in 2005 using the chi square test.

aged 15-19 years. The reduction in FGM prevalence in young women could be regarded as a proxy for declining incidence overall, as it is likely from our data set that the surgical procedures took place when they were children. These findings are in line with a recent report from 21 African countries showing in general a decreasing FGM prevalence over 30 years [7]. In the West African countries of Chad, Mali and Sierra Leone, however, the most severe form of FGM, infibulation, had increased by a few percentage points [7], and this was similar to what we found in Ethiopia. There was also a decreasing trend of FGM in the daughters of the mothers who were interviewed. This finding was striking in the daughters of mothers who had not undergone the procedure themselves where FGM prevalence in 2016 was less than 2%.

Second, opinions significantly changed over the time period with less than 20% of women interviewed in 2016 believing that the practice of FGM should continue. The changing opinion about FGM in Ethiopia was similar to that reported from Egypt where the proportion of ever-married women who believed FGM should be stopped increased from 14% in 1999 to 31% in 2014 [21].

Third, most women in our study in 2016 were circumcised as children, with nearly half having the procedure done as a young child aged 4 years or below, and the surgery was largely done by a non-medical traditional circumciser or birth attendant. This is similar to what has been reported from other African countries where the surgery is also largely conducted by non-medical practitioners [22–24]. This might explain the high incidence of immediate complications such as excessive bleeding, urine retention and swelling of the genital organs which was documented from interviewing women from selected districts of Bale Zone, Ethiopia [25].

In our study we did not obtain data on immediate or long-term complications. However, FGM is a practice associated with adverse health outcomes, such as poor mental health, psychological and emotional disturbance and persistent lower urinary tract symptoms, bacterial vaginosis and dyspareunia, all of which need better documentation, appropriate treatment and further research [3, 26, 27]. As discussed earlier, FGM is also associated with adverse maternal outcomes such as obstructed labour, post-partum bleeding and the need for emergency caesarean section [9].

Fourth, we identified and reaffirmed several factors associated with FGM. There were certain regions in the country with a high FGM prevalence of >75% (Dire Dawa, Harari and Affar) and in the Somali region FGM prevalence was >95%. In both Harari and Somali, there was no significant decline in FGM prevalence over the 16 years. Reasons for no progress in

**Table 5. Factors associated with female genital mutilation (FGM) in Ethiopia: 2016.**

| Characteristics | Interviewed | With FGM | | Pr (95% CI) | P value |
|---|---|---|---|---|---|
| | N | N | (%) | | |
| **Region**: | | | | | |
| Tigray | 540 | 131 | (24.2) | 1.0 | |
| Gambela | 21 | 7 | (33.3) | 1.4 (0.7-2.6) | 0.34 |
| SNNPR | 1652 | 1024 | (61.9) | 2.6 (2.2-3.0) | <0.001 |
| Benshagul-Gumuz | 75 | 47 | (62.7) | 2.6 (2.1-3.3) | <0.001 |
| Amhara | 1827 | 1127 | (61.7) | 2.5 (2.2-3.0) | <0.001 |
| Addis Ababa | 466 | 251 | (53.9) | 2.2 (1.9-2.6) | <0.001 |
| Oromiya | 2881 | 2178 | (75.6) | 3.1 (2.7-3.6) | <0.001 |
| Dire Dawa | 47 | 35 | (74.5) | 3.1 (2.5-3.8) | <0.001 |
| Harari | 18 | 15 | (83.3) | 3.4 (2.7-4.4) | <0.001 |
| Affar | 67 | 61 | (91.0) | 3.8 (3.2-4.4) | <0.001 |
| Somali | 228 | 225 | (98.7) | 4.1 (3.5-4.7) | <0.001 |
| **Education**: | | | | | |
| None | 3787 | 2758 | (72.8) | 1.0 | |
| Primary level | 2679 | 1662 | (62.0) | 0.8 (0.8-0.9) | <0.001 |
| Secondary level | 907 | 453 | (49.9) | 0.7 (0.6-0.7) | <0.001 |
| Tertiary level | 450 | 228 | (50.7) | 0.7 (0.6-0.8) | <0.001 |
| **Religion**: | | | | | |
| Traditional | 61 | 33 | (54.8) | 1.0 | |
| Catholic | 66 | 39 | (58.2) | 1.1 (0.8-1.5) | 0.57 |
| Protestant | 1862 | 1226 | (65.8) | 1.2 (0.9-1.5) | 0.06 |
| Orthodox Christian | 3424 | 1856 | (54.2) | 1.0 (0.8-1.3) | 0.98 |
| Muslim | 2363 | 1942 | (82.2) | 1.5 (1.2-1.9) | <0.001 |
| Other | 46 | 5 | (10.8) | 0.2 (0.1-0.5) | <0.001 |
| **Residence**: | | | | | |
| Urban | 1714 | 924 | (53.9) | 1.0 | |
| Rural | 6108 | 4177 | (68.4) | 1.3 91.2-1.3) | <0.001 |
| **Occupation**: | | | | | |
| Unemployed | 3839 | 2564 | (66.8) | 1.0 | |
| Professional | 1729 | 1138 | (65.8) | 0.9 (0.9-1.0) | 0.47 |
| Agricultural | 1629 | 1055 | (64.8) | 0.9 (0.9-1.0) | 0.15 |
| Manual | 441 | 242 | (54.9) | 0.8 (0.7-0.9) | <0.001 |
| Other | 184 | 102 | (55.4) | 0.8 (0.7-0.9) | <0.05 |
| **Wealth Index***: | | | | | |
| Poorest | 1306 | 852 | (65.2) | 1.2 (1.1-1.2) | <0.001 |
| Poorer | 1419 | 939 | (66.2) | 1.2 (1.1-1.2) | <0.001 |
| Middle | 1520 | 1086 | (71.4) | 1.3 (1.2-1.3) | <0.001 |
| Richer | 1529 | 1063 | (69.5) | 1.2 91.2-1.3) | <0.001 |
| Richest | 2048 | 1161 | (56.7) | 1.0 | |

FGM = female genital mutilation; SNNPR = Southern Nations Nationalities and Peoples Region; Pr = prevalence ratio; CI = confidence intervals.

*Wealth index refers to household wealth index as described in reference 16.

All data were weighted according to DHS guidance [20].

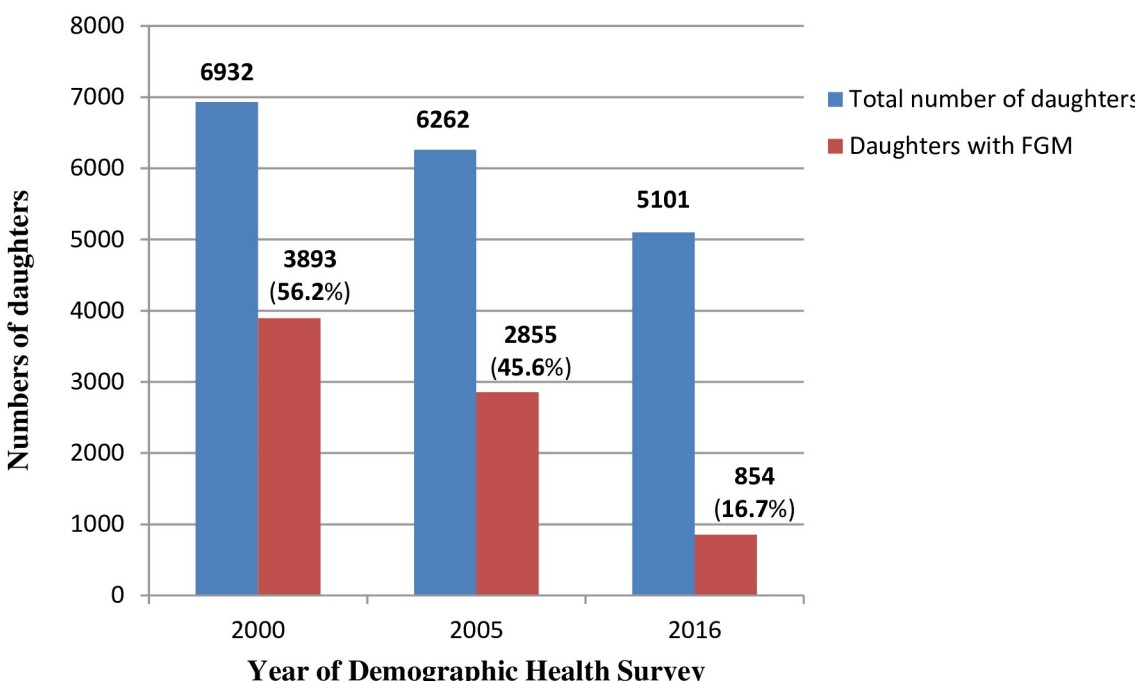

**Fig 2. Female genital mutilation (FGM) in daughters born to mothers who had FGM in Ethiopia in 2000, 2005 and 2016.**

FGM prevalence in Somali and Harari regions may relate to ethnicity and long-held traditional beliefs that circumcision is a necessary prerequisite for marriageability [28]. Geographic variation as observed in Ethiopia is not uncommon, with Kenya reporting similar findings several years ago [29].

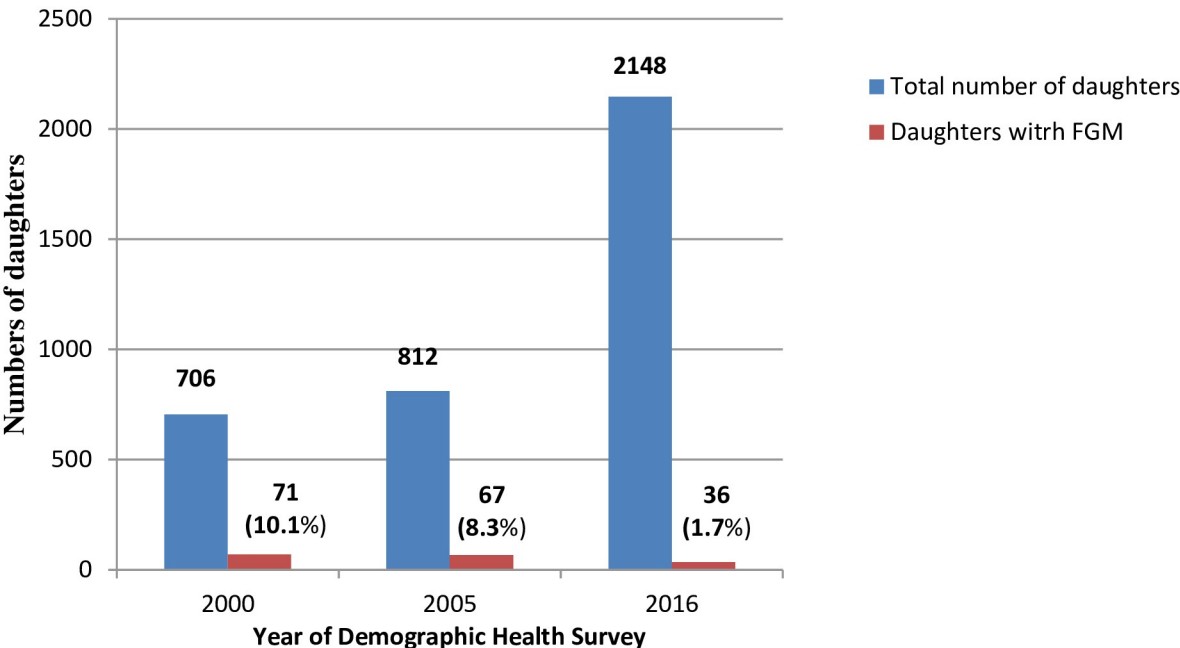

**Fig 3. Female genital mutilation (FGM) in daughters born to mothers who did not have FGM in Ethiopia in 2000, 2005 and 2016.**

Any school education from primary up to tertiary level appeared to be protective of FGM compared with no education, similar to the report from Bale Zone, Ethiopia [30] and recent findings from Egypt and Nigeria [21, 31]. The important role of education is further emphasised by an Ethiopian study showing that school-based awareness campaigns result in young children wanting to abandon the practice of FGM [32]. Muslim women in our study had a high but decreasing prevalence of FGM, similar to DHS findings in Kenya [33], but different to Nigeria where FGM prevalence was low in women of Islamic faith [31]. Women of Catholic and Protestant denominations had high FGM prevalence which did decrease from 2005 to 2016. The reasons for this are unclear and differ from what has been reported in neighbouring Kenya [33]. The higher FGM prevalence in rural compared with urban areas aligns with findings from Kenya [33], but differs from what was reported from Nigeria [31]. Finally, the higher FGM prevalence in women from middle and poorer household wealth indexes accords exactly with findings from Egypt [21].

The strengths of this study were the national coverage using data from EDHS, the representativeness of the study sample, the weighting of data before analysis to compensate for the unequal probability of selection amongst the regions, the same standardised questions about socio-demographic factors and FGM being asked at each of the three time points and the conduct and reporting of the study which followed international guidelines (STROBE—Strengthening the Reporting of Observational Studies in Epidemiology) [34].

There were however some limitations. In contrast to the 2000 and 2016 EDHS, in 2005 there were some "do not know" responses for the sociodemographic variables related to FGM, thus preventing the full data set for that year from being analysed. There is also the important question of whether self-reported data on FGM is reliable. While two studies have indicated that self-reported FGM is a reasonably reliable proxy measurement of FGM prevalence [13, 35], a study from rural Tanzania showed that reported FGM prevalence underestimated the true extent of circumcision as determined clinically [36].

We used the same methods on analysing the 2000, 2005 and 2016 data sets and as recommended, we weighted the actual data that was downloaded from DHS Measure web site.

There are two main programmatic implications from our study. First, the study highlights the regions of Ethiopia (Somali, Affar and Harari) where progress in eliminating FGM is either slow or stagnant. In Somali and Harari regions, while marriageability is cited amongst women as a major reason for continuing FGM, young men contradict this opinion and would prefer to marry uncircumcised girls [28]. In these regions and in other rural areas, targeted integrated interventions involving mothers, young men, religious leaders and schools should be undertaken to educate and empower women and men about the importance of discontinuing FGM. A recent study in Egypt suggests that policies to empower women through education and improved income could work in reducing the on-going practice of FGM [37]. Older individuals in Ethiopia hold the strongest views in favour of FGM, and there is also great potential here for educating and persuading this group about the importance of eliminating the practice [38]. Targeting fathers in education campaigns should not be forgotten as studies in Ethiopia and neighbouring Sudan show that they are important decision makers about FGM [39, 40]. Thus, a concerted effort is needed to empower women through education and wealth and to target and better educate groups who still strongly believe in the value of FGM. Second, it is critically important to ensure that all girls get some form of education, not only for the wider social, economic and health benefits that this brings, but because in this context it has been shown to be so important as a protective factor against FGM.

## Conclusion

We used data from three EDHS conducted in 2000, 2005 and 2016 to assess changes in prevalence and self-reported practices for FGM among women of reproductive age and their daughters. Although progress has been slow, the prevalence of FGM amongst women, especially younger women, and their daughters declined over time and an increasing proportion of women felt that the practice of FGM should stop. Factors associated with higher FGM prevalence and lack of progress included living in certain regions, especially the Somali region, lack of school education, coming from rural rather than urban areas, and living in less wealthy households. Recommendations for moving forward include empowering women through education and wealth, targeting integrated interventions to mothers, fathers, youngsters, religious leaders and schools and ensuring that all girls receive some form of education.

## Acknowledgments

This research was conducted through the Structured Operational Research and Training Initiative (SORT IT), a global partnership led by the Special Programme for Research and Training in Tropical Diseases at the World Health Organization (WHO/TDR). The model is based on a course developed jointly by the International Union Against Tuberculosis and Lung Disease (The Union) and Medécins sans Frontières (MSF/Doctors Without Borders). The specific SORT IT programme which resulted in this publication was jointly organised, implemented and mentored by the Centre for Operational Research, The Union, Paris, France; MSF-Luxembourg (MSF LuxOR); MSF-Belgium (MSF-OCB); the University of Bergen, Norway and the London School of Hygiene and Tropical Medicine, London, UK.

## Author Contributions

**Conceptualization:** Gedion Asnake Azeze, Anita Williams, Hannock Tweya, Anthony D. Harries.

**Data curation:** Gedion Asnake Azeze.

**Formal analysis:** Gedion Asnake Azeze, Anita Williams, Hannock Tweya, Mohammed Suleiman Obsa, Taklu Marama Mokonnon, Zewde Zema Kanche, Robera Olana Fite, Anthony D. Harries.

**Funding acquisition:** Anita Williams, Anthony D. Harries.

**Methodology:** Gedion Asnake Azeze, Anita Williams, Hannock Tweya, Anthony D. Harries.

**Software:** Gedion Asnake Azeze.

**Supervision:** Anthony D. Harries.

**Writing – original draft:** Gedion Asnake Azeze, Anthony D. Harries.

**Writing – review & editing:** Gedion Asnake Azeze, Anthony D. Harries.

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
