## [Decision Letter · Decision Letter 0]

25 May 2020

PONE-D-20-05281

Changing prevalence and factors associated with Female Genital Mutilation in Ethiopia: data from the 2000, 2005 and 2016 National Demographic Health Surveys.

PLOS ONE

Dear Dr. Azeze,

Thank you for submitting your manuscript to PLOS ONE. After careful consideration, we feel that it has merit but does not fully meet PLOS ONE’s publication criteria as it currently stands. Therefore, we invite you to submit a revised version of the manuscript that addresses the points raised during the review process.

Please respond in detail to the comments from both reviewers in your updated draft, including updated analysis methods. Thank you.

We look forward to receiving your revised manuscript.

Kind regards,

Amy Michelle DeBaets, PhD

Academic Editor

PLOS ONE

Journal Requirements:

3. Your ethics statement must appear in the Methods section of your manuscript. If your ethics statement is written in any section besides the Methods, please move it to the Methods section and delete it from any other section. Please also ensure that your ethics statement is included in your manuscript, as the ethics section of your online submission will not be published alongside your manuscript.

Reviewers' comments:

Reviewer's Responses to Questions

**Comments to the Author**

1. Is the manuscript technically sound, and do the data support the conclusions?

Reviewer #1: Yes

Reviewer #2: No

2. Has the statistical analysis been performed appropriately and rigorously? 

Reviewer #1: Yes

Reviewer #2: No

3. Have the authors made all data underlying the findings in their manuscript fully available?

Reviewer #1: Yes

Reviewer #2: Yes

4. Is the manuscript presented in an intelligible fashion and written in standard English?

Reviewer #1: Yes

Reviewer #2: Yes

5. Review Comments to the Author

Reviewer #1: The author intended to show how Female Genital Mutilation decrease overtime by using the DHS data of the 2000, 2005 and 2016.

FGM is the harmful traditional practice which was a worldwide problem and a major concern for Africa.

Ethiopia is also one of the country where FGM is highly prevalent. All regions in Ethiopia practiced FGM, and the prevalence ranges from lowest (Tigri) to the highest (Somali). Although the DHS data of 2000, 2005 and 2016 showed a decrease in the prevalence of FGM in Ethiopia, due to its deep-rooted nature, the practice still continued and needs further work.

The study tried to show the decreased prevalence and some factors associated with the practice. I am happy to read this interesting article. However, as a reviewer the author should take into consideration the following comments.

Abstract

Setting: the explanation is definition better to change to Introduction

Female genital mutilation is defined clearly by WHO

The traditional surgical modification might hide the actual definition because modification when it comes to FGM is about reconstructive surgery or any surgery which modified the injured part of the genitalia, so better to use the WHO definition.

Objective

State only the objective (what you intended to study) (30-33)

Introduction

FGM in Ethiopia,

Who classified FGM in to four (avoid incomplete sentence)

Conceder the type of FGM practiced most in Ethiopia (type1&2) (60)

You mentioned the problem in short as worldwide and Africa. Better to state some points how the problem is huge in Africa.

The problem in Ethiopia

Better to mention those who practice FGM such as the regions, ethnicity and religion before mentioning the least and the highest (82,83)

Methodology

Setting

Better to say something about the mortality rate, the literacy status of women and the harmful traditional practice commonly practice including FGM

Although the study showed the declining trend of the practice, girls and women are still victimized, so try to make your conclusion and recommendation strong.

Reviewer #2: The authors used three recent Demographic and Health Surveys (DHS) conducted in Ethiopia to describe trends in female genital cutting (FGC) in the country over time. This is a topic of clear importance to women’s wellbeing in Ethiopia and elsewhere and I am glad to see Ethiopian researchers addressing it. However, there are problems with the statistics presented in this article that must be corrected before it is suitable for publication. Comments specific to each section of the manuscript follow.

Introduction

I would like to see the authors acknowledge that the health consequences of FGC likely vary with the severity of the procedure. The review by Klein et al. (number 3) that is cited is of poor quality. (For example, no recognized methods of bias assessment were used.) I suggest citing a slightly older but more methodologically rigorous review by Berg et al. published in BMJ Open in 2014 instead of the current citation, or at least in addition to it.

Methods

General information on Ethiopia such as its geographic location within Africa, life expectancy, GDP, etc. is unnecessary and does not directly inform the analyses described in this paper. I recommend that the authors remove it.

Analysis and statistics

The use of sampling weights is necessary for nearly all analyses of DHS data and the authors are clearly aware of that need. However, I am not convinced that their use of the weighting variables was correct for two reasons. First, the proportions indicated in the results appear to be unweighted. For example, the precise prevalence of FGC in 2016 that the authors report in Figure 1 can be obtained from dividing the unweighted number of women who underwent FGC by the total number of women interviewed (5101/7248 = 70.4%). This would probably not be the case if weighting was properly done. Second, the prevalence of FGC in 2016 reported by the authors (70.4%) is notably different from that reported in the 2016 Ethiopia DHS final report available on the DHS website, which reports that the weighted prevalence was 65%. Similar problems arise when looking at region-specific estimates. The authors estimate that 28.4% of women in Tigray had undergone FGC in 2016, but the official DHS report from that year indicates that the proportion was 24%.

The citation that the authors provide for their weighting approach (number 20) is for an online forum that discusses weighting for specific sub-samples of the DHS. Since the authors evaluate the entire DHS sample of women (15-49-year-olds), specific weights for subsamples should not have been necessary. More detailed information on how to use the weighting variables is provided directly from the DHS in their report titled Guide to DHS Statistics. If the authors believe that their estimates are correct they must discuss why they deviate from the estimates included in official DHS reports for the country. The issue of weighting must be addressed before the article is ready for publication.

I am also concerned about how the chi squared tests for trend were conducted. The authors do not indicate that they pooled all three datasets together, which leads me to believe that each dataset was analyzed independently. If that is the case, it is difficult to understand how chi squared tests for trend across all three years could have been conducted because this test requires information on the numerator and denominator used to generate the proportion within each sample. More details on the methodology are needed here. If helpful, the BMJ has published a clear text on the chi squared test for trend in its publication ‘Statistics at Square One’ (see Chapter 8).

All of the descriptive statistics presented should be accompanied by confidence intervals, including the proportions.

Results

I am not sure what went wrong during the estimation of odds ratios but some of the estimates presented are nonsensical. For example, the authors report an unadjusted odds ratio of 180.6 for Somali region when compared to Tigray. Using the numbers included in Table 5 we can set up an epidemiologic 2x2 table and calculate the odds of FGC in Tigray (131/462) and the odds of FGC in Somali (225/228) by hand. The unadjusted odds ratio comparing Somali to Tigray is therefore (225/228) / (131/462) = 3.5.

Adjusting the odds ratios is unnecessary in this study. Confounding would be a concern if the authors were interested in estimating the causal effect of each of these traits (i.e. living in a rural area, religion) on FGC status but that is not the case here. I recommend that these analyses be omitted entirely.

Discussion

I was surprised to see the authors state that “it is estimated that one in every 500 circumcisions results in death” (line 294) because I have never come across reliable estimates of mortality associated with FGC, so I followed up the reference. It appears that the reference cited (again from the Klein et al. study) simply took this claim from another commentary (not a scientific investigation) by Marcel Reyners (Reviews in Gynaecological Practice, 2004), who based his estimate on a single source from Egypt in 1989. I was unable to locate the original Egyptian source but even so, it is clear that this estimate is not well supported by evidence and should not be included in a scientific article.

The authors acknowledge and discuss the reliability of self-reports of FGC status and severity. However, they state in lines 328-330 that “All three studies showed that there was no correlation between DHS FGM responses and anatomic extent of cutting.” Only one of the three articles cited (number 35, Bjalkander et al.) actually compares clinically observed FGC severity with DHS responses. Unfortunately, the Bjalkander study is misguided because the authors compared nationally representative DHS statistics from Sierra Leone with the distribution of FGC types observed in a sample taken from one part of the country. (This is concerning because, as the authors highlight in Ethiopia, there are often dramatic differences in FGC prevalence and type between regions.) These references need to be more carefully assessed and the implication that there is no correlation between DHS responses and the actual extent of cutting should be omitted because it is not sufficiently supported by evidence. Moreover, please use caution: the statement noted above from lines 328-330 is nearly a direct quote from the abstract of the Bjalkander article.

6. PLOS authors have the option to publish the peer review history of their article (what does this mean?). If published, this will include your full peer review and any attached files.

Reviewer #1: No

Reviewer #2: Yes: Alissa Koski

---

## [Author Response · Author response to Decision Letter 0]

16 Jun 2020

Amy Michelle DeBaets, PhD

Academic Editor

PLOS ONE

Dear Dr DeBaets, 

Thank you for considering our manuscript and for arranging for it to be reviewed by two reviewers. We have tried to address your comments and the comments / suggestions from the two reviewers. In the Response to Reviewers, we copy each of the comments and provide the RESPONSE underneath. We also provide a marked-up copy of the manuscript that highlights changes made to the original version and this is uploaded as a separate file called “Revised Manuscript with Track Changes”. Finally, we provide an unmarked version of the revised manuscript without tracked changes and this is uploaded as a separate file labelled 'Manuscript'. We remade all the tables in red font rather than tracking as tracking was so messy and we could not see what we were doing.

We hope we have satisfactorily addressed al the comments and hope that our paper may now be suitable for publication in your journal. 

Best wishes 

Gedion Asnake on behalf of the co-authors

Editor: 

RESPONSE:

We have checked and have ensured that the manuscript meets the journal’s requirements. 

We note that you have indicated that data from this study are available upon request. PLOS only allows data to be available upon request if there are legal or ethical restrictions on sharing data publicly. 

RESPONSE:

The principal investigator of this study was authorized to download the survey data from the Demographic and Health Surveys (DHS) Program up on request for this registered study. The letter to the PI stated that the DHS data must not be passed on to others, without the written consent of DHS. However, to replicate our study findings, the DHS data are in the public domain on the DHS measure survey web site which is available at: https://dhsprogram.com/data/available-datasets.cfm

We have also made change on this in ‘Manuscript with Track Changes’. We have confirmed that the authors had no special access privileges to the DHS data, and other researchers will be able to access the data in the same manner as the authors using the provided URL link. Data access requests may also be sent to Bridgette Wellington (Data Archivist at The Demographic and Health Surveys (DHS) Program) at E-mail address: archive@dhsprogram.com We have modified ‘Availability of data’ accordingly.

Your ethics statement must appear in the Methods section of your manuscript. If your ethics statement is written in any section besides the Methods, please move it to the Methods section and delete it from any other section. Please also ensure that your ethics statement is included in your manuscript, as the ethics section of your online submission will not be published alongside your manuscript.

RESPONSE:

The ethic statement is already in the Methods on lines 187-194. However, it was also in the Declarations and we have removed it from here. 

Reviewer #1: 

The author intended to show how Female Genital Mutilation decreases overtime by using the DHS data of the 2000, 2005 and 2016. FGM is the harmful traditional practice which was a worldwide problem and a major concern for Africa. Ethiopia is also one of the countries where FGM is highly prevalent. All regions in Ethiopia practiced FGM, and the prevalence ranges from lowest (Tigri) to the highest (Somali). Although the DHS data of 2000, 2005 and 2016 showed a decrease in the prevalence of FGM in Ethiopia, due to its deep-rooted nature, the practice still continued and needs further work. The study tried to show the decreased prevalence and some factors associated with the practice. I am happy to read this interesting article. 

RESPONSE: 

Thank you for the encouraging response

Abstract

Abstract Setting: the explanation is definition better to change to Introduction

Female genital mutilation is defined clearly by WHO. The traditional surgical modification might hide the actual definition because modification when it comes to FGM is about reconstructive surgery or any surgery which modified the injured part of the genitalia, so better to use the WHO definition.

RESPONSE:

Thank you. We have now included the WHO definition of FGM. This has taken us over the word count limit and we have therefore had to make some other changes to get us back to the word count limit. 

Abstract Objective

State only the objective (what you intended to study) (30-33)

RESPONSE:

Thank you. We have changed the sentence in the Abstract section on Objective

Introduction

FGM in Ethiopia,

WHO classified FGM in to four (avoid incomplete sentence). Consider the type of FGM practiced most in Ethiopia (type1&2) (60)

RESPONSE:

Thank you. We have expanded on the different types of FGM and explained that Types I and II are the most common forms. 

You mentioned the problem in short as worldwide and Africa. Better to state some points how the problem is huge in Africa.

RESPONSE:

Thank you. We have expanded on some of the points that make FGM a common practice in Africa. 

The problem in Ethiopia

Better to mention those who practice FGM such as the regions, ethnicity and religion before mentioning the least and the highest (82,83)

RESPONSE:

Thank you. We discuss at length the issues of region, ethnicity, religion, education and wealth in the Discussion in relation to our study findings. We would prefer to just mention the prevalence of FGM in the country as a whole and give the prevalence ranges from the lowest to the highest in the different regions. The Introduction is already quite long at 2.5 pages and we prefer not to increase it further at this stage. We hope the Editor and the reviewer can accept our stance here. 

Methodology

Setting

Better to say something about the mortality rate, the literacy status of women and the harmful traditional practice commonly practice including FGM

RESPONSE:

Good point, thank you. We have included under Setting, information about female literacy, use of traditional practices, maternal mortality ratio and under five mortality rates. 

Although the study showed the declining trend of the practice, girls and women are still victimized, so try to make your conclusion and recommendation strong.

RESPONSE:

Thank you. We have strengthened the recommendations in the final paragraph of the Discussion and in the Conclusion. 

Reviewer #2: 

The authors used three recent Demographic and Health Surveys (DHS) conducted in Ethiopia to describe trends in female genital cutting (FGC) in the country over time. This is a topic of clear importance to women’s wellbeing in Ethiopia and elsewhere and I am glad to see Ethiopian researchers addressing it. However, there are problems with the statistics presented in this article that must be corrected before it is suitable for publication. Comments specific to each section of the manuscript follow.

RESPONSE:

Thank you for the encouraging comment. We will take note of your advice about statistics. 

Introduction

I would like to see the authors acknowledge that the health consequences of FGC likely vary with the severity of the procedure. The review by Klein et al. (number 3) that is cited is of poor quality. (For example, no recognized methods of bias assessment were used.) I suggest citing a slightly older but more methodologically rigorous review by Berg et al. published in BMJ Open in 2014 instead of the current citation, or at least in addition to it.

RESPONSE:

Thank you for this helpful and better reference. We have replaced Klein et al with Berg RC et al as reference 3. We have expanded on the health consequences and the fact that they relate to severity of the procedure. 

Methods

General information on Ethiopia such as its geographic location within Africa, life expectancy, GDP, etc. is unnecessary and does not directly inform the analyses described in this paper. I recommend that the authors remove it.

RESPONSE:

Thank you. Reviewer 1, however, wanted us to expand this section and present data on use of traditional practices, female literacy and maternal and under five mortality. We therefore hope this reviewer can accept that we keep the section with some extra information added. For readers who do not know Ethiopia, this may also be helpful. 

Analysis and statistics

The use of sampling weights is necessary for nearly all analyses of DHS data and the authors are clearly aware of that need. However, I am not convinced that their use of the weighting variables was correct for two reasons. First, the proportions indicated in the results appear to be unweighted. For example, the precise prevalence of FGC in 2016 that the authors report in Figure 1 can be obtained from dividing the unweighted number of women who underwent FGC by the total number of women interviewed (5101/7248 = 70.4%). This would probably not be the case if weighting was properly done. Second, the prevalence of FGC in 2016 reported by the authors (70.4%) is notably different from that reported in the 2016 Ethiopia DHS final report available on the DHS website, which reports that the weighted prevalence was 65%. Similar problems arise when looking at region-specific estimates. The authors estimate that 28.4% of women in Tigray had undergone FGC in 2016, but the official DHS report from that year indicates that the proportion was 24%.

The citation that the authors provide for their weighting approach (number 20) is for an online forum that discusses weighting for specific sub-samples of the DHS. Since the authors evaluate the entire DHS sample of women (15-49-year-olds), specific weights for subsamples should not have been necessary. More detailed information on how to use the weighting variables is provided directly from the DHS in their report titled Guide to DHS Statistics. If the authors believe that their estimates are correct they must discuss why they deviate from the estimates included in official DHS reports for the country. The issue of weighting must be addressed before the article is ready for publication.

RESPONSE:

Thank you. 

In the original submission, the weighted prevalence for 2016 was calculated in the same way as for 2000 and 2005. This was done first by computing (generating) a new variable 'WGT' by dividing an already existing variable (that is 'V005') by one million (1000000). That is "COMPUTE WGT = V005/1000000.". We then turned on the newly created variable 'WGT' to apply itself automatically to every tabulation: we applied the weight by writing the following command on the syntax file "WEIGHT BY WGT." We used the same method to weight the three datasets. We found that our weighted prevalence of FGM for 2000 (79.9%) and 2005 (74.3%) matched the findings of the DHS datasets for the official final 2000 and 2005 EDHS reports respectively. Although we used the same method to weight the three DHS dataset, our finding on weighted prevalence of FGM for 2016 in the DHS dataset (70.4%) didn't match with that of the official final EDHS 2016 report, which was 65.2%. 

Following your query, we tried to discuss the issue on the online forum 'The DHS Program User Forum'. After a week, we received a message from a DHS analysis expert with the following direction: " You need to include in the denominator all of the cases for which the questions on FGC were asked. In this survey g102 is missing if g100 is not Yes. Also, in this survey FGC was asked in a subsample, but you can select that subsample based on variable g100. Try the following: if (not sysmis(g100)) fgc = 0; if (g102 = 1) fgc = 1; then weight the data as you have for other surveys, and you should match the estimate in the report." 

The discussion with the DHS analysis experts on an online forum is found on the following URL link under the section country 'Ethiopia' https://userforum.dhsprogram.com/index.php?t=msg&th=8082&start=0& . 

Using the above suggested command by DHS analysis experts, we then created a new variable 'fgc' in our dataset. Then we weighted the 2016 dataset before running any tabulation, which is what we did for the 2000 and 2005 dataset. When we ran the frequency analysis for the newly created variable 'fgc', FGM prevalence became 65.2%. The denominator (the total number of women interviewed) became 7822 while the numerator (total number of women circumcised) was unchanged at 5101. 

Similarly, problems at region-specific estimates were solved. Thank you for helping us to correct this and ensure our presented data for 2016 now match the 2016 EDHS report.

I am also concerned about how the chi squared tests for trend were conducted. The authors do not indicate that they pooled all three datasets together, which leads me to believe that each dataset was analyzed independently. If that is the case, it is difficult to understand how chi squared tests for trend across all three years could have been conducted because this test requires information on the numerator and denominator used to generate the proportion within each sample. More details on the methodology are needed here. If helpful, the BMJ has published a clear text on the chi squared test for trend in its publication ‘Statistics at Square One’ (see Chapter 8).

RESPONSE:

We apologise for the misuse of the chi square test for trend. We should have just compared the proportions (numerator over denominator) with each other in each year. We have redone these calculations and have now compared proportions in 2005 with 2000 and proportions in 2016 with 2005. The tables (1-4) have been modified to show this. As the tables have been changed in a major way we have not done this in tracking and just present the new tables in red font: we hope this is acceptable. 

All of the descriptive statistics presented should be accompanied by confidence intervals, including the proportions.

RESPONSE

Thank you. We have now presented proportions with 95% confidence intervals in Tables 1, 2 and 3. However, Table 4 is already very busy and presenting 95% confidence intervals here we believe overloads the table and makes it difficult to read. In Table 4 we prefer to just keep the percentages without the confidence intervals. We have done the same with Table 5 which also shows Prevalence ratios (Pr). We hope you can accept our stance here. 

Results

I am not sure what went wrong during the estimation of odds ratios but some of the estimates presented are nonsensical. For example, the authors report an unadjusted odds ratio of 180.6 for Somali region when compared to Tigray. Using the numbers included in Table 5 we can set up an epidemiologic 2x2 table and calculate the odds of FGC in Tigray (131/462) and the odds of FGC in Somali (225/228) by hand. The unadjusted odds ratio comparing Somali to Tigray is therefore (225/228) / (131/462) = 3.5. Adjusting the odds ratios is unnecessary in this study. Confounding would be a concern if the authors were interested in estimating the causal effect of each of these traits (i.e. living in a rural area, religion) on FGC status but that is not the case here. I recommend that these analyses be omitted entirely.

RESPONSE:

Thank you. We have made some substantial changes to Table 5. First, we have moved from odds ratios to prevalence ratios as we believe these are easier to understand for the reader. Second, you are right and we have removed the adjusted analyses and just present the crude Prevalence ratios (Pr). We have modified the Results narrative around Table 5 to account for these slightly different results. 

Discussion

I was surprised to see the authors state that “it is estimated that one in every 500 circumcisions results in death” (line 294) because I have never come across reliable estimates of mortality associated with FGC, so I followed up the reference. It appears that the reference cited (again from the Klein et al. study) simply took this claim from another commentary (not a scientific investigation) by Marcel Reyners (Reviews in Gynaecological Practice, 2004), who based his estimate on a single source from Egypt in 1989. I was unable to locate the original Egyptian source but even so, it is clear that this estimate is not well supported by evidence and should not be included in a scientific article.

RESPONSE:

Thank you for this. We have removed it. 

The authors acknowledge and discuss the reliability of self-reports of FGC status and severity. However, they state in lines 328-330 that “All three studies showed that there was no correlation between DHS FGM responses and anatomic extent of cutting.” Only one of the three articles cited (number 35, Bjalkander et al.) actually compares clinically observed FGC severity with DHS responses. Unfortunately, the Bjalkander study is misguided because the authors compared nationally representative DHS statistics from Sierra Leone with the distribution of FGC types observed in a sample taken from one part of the country. (This is concerning because, as the authors highlight in Ethiopia, there are often dramatic differences in FGC prevalence and type between regions.) These references need to be more carefully assessed and the implication that there is no correlation between DHS responses and the actual extent of cutting should be omitted because it is not sufficiently supported by evidence. Moreover, please use caution: the statement noted above from lines 328-330 is nearly a direct quote from the abstract of the Bjalkander article.

RESPONSE:

Thank you for this good advice. We have decided to remove this part of the limitations about FGM responses and extent of cutting.

---

## [Decision Letter · Decision Letter 1]

4 Aug 2020

PONE-D-20-05281R1

Changing prevalence and factors associated with Female Genital Mutilation in Ethiopia: data from the 2000, 2005 and 2016 National Demographic Health Surveys.

PLOS ONE

Dear Dr. Azeze,

Thank you for submitting your manuscript to PLOS ONE. After careful consideration, we feel that it has merit but does not fully meet PLOS ONE’s publication criteria as it currently stands. Therefore, we invite you to submit a revised version of the manuscript that addresses the points raised during the review process.

Please make the minor corrections requested by the reviewer below. We look forward to receiving the corrections and the opportunity to publish your article once they are made.

We look forward to receiving your revised manuscript.

Kind regards,

Amy Michelle DeBaets, PhD

Academic Editor

PLOS ONE

Reviewers' comments:

Reviewer's Responses to Questions

**Comments to the Author**

1. If the authors have adequately addressed your comments raised in a previous round of review and you feel that this manuscript is now acceptable for publication, you may indicate that here to bypass the “Comments to the Author” section, enter your conflict of interest statement in the “Confidential to Editor” section, and submit your "Accept" recommendation.

Reviewer #2: (No Response)

2. Is the manuscript technically sound, and do the data support the conclusions?

Reviewer #2: Yes

3. Has the statistical analysis been performed appropriately and rigorously? 

Reviewer #2: Yes

4. Have the authors made all data underlying the findings in their manuscript fully available?

Reviewer #2: Yes

5. Is the manuscript presented in an intelligible fashion and written in standard English?

Reviewer #2: Yes

6. Review Comments to the Author

Reviewer #2: The authors responded to all of the concerns I raised in my initial review sufficiently. However, the abstract has not been updated to reflect changes to their methods and results: the prevalence estimate for 2016 does not correspond with what is reported later in the paper (line 195) and the methods still refer to regression adjustment for confounding. This must be corrected.

7. PLOS authors have the option to publish the peer review history of their article (what does this mean?). If published, this will include your full peer review and any attached files.

Reviewer #2: **Yes: **Alissa Koski

---

## [Author Response · Author response to Decision Letter 1]

12 Aug 2020

Amy Michelle DeBaets, PhD

Academic Editor

PLOS ONE

Dear Dr DeBaets, 

Once again, we would like to thank you for considering our manuscript and for arranging for it to be reviewed by reviewer(s). We have tried to address your comments and the comments / suggestions from the reviewer(s). In the ‘Response to Reviewers’, we copy each of the comments and provide the RESPONSE in bold underneath. We also provide a marked-up copy of the manuscript that highlights changes made to the original version and this is uploaded as a separate file called “Manuscript with Track Changes”. Finally, we provide an unmarked version of the revised manuscript without tracked changes and this is uploaded as a separate file labelled 'Manuscript'. 

We hope we have satisfactorily addressed all the comments and hope that our paper may now be suitable for publication in your journal. 

Best wishes 

Gedion Asnake on behalf of the co-authors

Academic Editor: 

While revising your submission, please upload your figure files to the Preflight Analysis and Conversion Engine (PACE) digital diagnostic tool, https://pacev2.apexcovantage.com/. PACE helps ensure that figures meet PLOS requirements. To use PACE, you must first register as a user. Registration is free. Then, login and navigate to the UPLOAD tab, where you will find detailed instructions on how to use the tool. 

RESPONSE:

We have uploaded our figure files to the Preflight Analysis and Conversion Engine (PACE) https://pacev2.apexcovantage.com/ digital diagnostic tool. We have downloaded the converted figure files separately and resubmitted together with the revised version of the manuscript.

Reviewer #2: 

The authors responded to all of the concerns I raised in my initial review sufficiently. However, the abstract has not been updated to reflect changes to their methods and results: the prevalence estimate for 2016 does not correspond with what is reported later in the paper (line 195) and the methods still refer to regression adjustment for confounding. This must be corrected. 

RESPONSE: 

Thank you for the encouraging response. We have updated the ‘Abstract’ section to reflect changes. On this ‘Abstract’ section, the prevalence estimate for 2016 (line 39) has been updated and now corresponds with what has been reported on the ‘Result’ section (line 195). We also modified the method section accordingly (line 36-38).

---

## [Editor Report · Decision Letter 2]

19 Aug 2020

Changing prevalence and factors associated with Female Genital Mutilation in Ethiopia: data from the 2000, 2005 and 2016 National Demographic Health Surveys.

PONE-D-20-05281R2

Dear Dr. Azeze,

We’re pleased to inform you that your manuscript has been judged scientifically suitable for publication and will be formally accepted for publication once it meets all outstanding technical requirements.

Kind regards,

Amy Michelle DeBaets, PhD

Academic Editor

PLOS ONE
---

## [Editor Report · Acceptance letter]

25 Aug 2020

PONE-D-20-05281R2 

Changing prevalence and factors associated with Female Genital Mutilation in Ethiopia: data from the 2000, 2005 and 2016 National Demographic Health Surveys. 

Dear Dr. Azeze:

I'm pleased to inform you that your manuscript has been deemed suitable for publication in PLOS ONE. Congratulations! Your manuscript is now with our production department. 

Kind regards, 

on behalf of

Dr. Amy Michelle DeBaets 

Academic Editor

PLOS ONE